# Breastfeeding, cognitive ability, and residual confounding: A comment on studies by Pereyra-Elìas et al.

**Kimmo Sorjonen**[1]*, **Gustav Nilsonne**[1,2], **Michael Ingre**[3,4], **Bo Melin**[1]

**1** Department of Clinical Neuroscience, Karolinska Institutet, Stockholm, Sweden, **2** Department of Psychology, Stockholm University, Stockholm, Sweden, **3** Department of Gastroenterology and Hepatology, Karolinska University Hospital Huddinge, Karolinska Institutet, Stockholm, Sweden, **4** Institute for Globally Distributed Open Research and Education (IGDORE), Stockholm, Sweden

* kimmo.sorjonen@ki.se

**Data Availability Statement:** The datasets used in this article are publicly available in the UK Data Service at http://doi.org/10.5255/UKDA-SN-8172-4 [Accessed 26 June 2023].

## Abstract

Recent studies found positive effects of breastfeeding on the child's cognitive ability and educational outcomes even when adjusting for maternal cognitive ability in addition to a large number of other potential confounders. The authors claimed an important role of breastfeeding for the child's cognitive scores. However, it is well known that error in the measurement of confounders can leave room for residual confounding. In the present reanalyses, we found incongruent effects indicating simultaneous increasing and decreasing effects of breastfeeding on the child's cognitive ability and educational outcomes. We conclude that findings in the reanalyses may have been due to residual confounding due to error in the measurement of maternal cognitive ability. Consequently, it appears premature to assume a genuine increasing effect of breastfeeding on the child's cognitive ability and educational outcomes and claims in this regard may be challenged.

## Introduction

Studies have reported a positive association between breastfeeding and the child's cognitive ability [1–4]. However, these estimated associations do not necessarily imply that breastfeeding has a genuine (non-spurious) effect on cognitive ability. The association could, for example, be confounded by maternal cognitive ability, because maternal cognitive ability appears to have a positive association with breastfeeding [4–6] and cognitive ability is strongly hereditary [7–10]. Sullivan et al. [11] reported a correlation of 0.35 between maternal and children's vocabulary scores. Studies adjusting for maternal cognitive ability have shown mixed results, some finding [2,4,5,12] and some not finding [13,14] a persisting positive association between breastfeeding and the child's cognitive ability. Recently, Pereyra-Elìas et al. [15] found associations to be greatly reduced when adjusting for maternal cognitive ability, in addition to socioeconomic position and other potential confounders, but they remained, for the most part, statistically significant. Pereyra-Elìas et al. [15] concluded that "the role of breastfeeding on the child's cognitive scores should not be underestimated" (p. 15). In the same population,

**Funding:** The authors received no specific funding for this work.

**Competing interests:** The authors have declared that no competing interests exist.

Pereyra-Elìas et al. [16] also found positive associations between breastfeeding and the child's educational outcomes that persisted when adjusting for maternal cognitive ability, socioeconomic position, and other potential confounders.

Although Pereyra-Elìas et al. [15,16] adjusted for a large number of possible confounders, e.g. socioeconomic position and maternal cognitive ability, they did not appear to consider or discuss the possible impact of measurement error. An often overlooked assumption in regression analysis is that predictors are measured without error, i.e. with perfect reliability [17]. It has been shown that with measurement error in a confounder Z, the effect of a predictor X on an outcome Y may remain statistically significant when adjusting for Z, even if Z would fully account for the association between X and Y if not measured with error [18–31].

We have previously estimated the effect of breastfeeding on intergenerational change in cognitive ability, from mothers to their children but also backward from children to their mothers, with latent change score modeling. Our findings were contradictory, indicating simultaneous increasing and decreasing effects of breastfeeding on intergenerational change in cognitive ability. We concluded that the findings were spurious, probably due to residual confounding and regression to the mean [32].

The objective of the present study was to reanalyze the data used by Pereyra-Elìas et al. [15,16] and to evaluate their conclusion that a positive effect of breastfeeding on the child's cognitive ability and educational outcomes persisted when adjusting for possible confounders, including maternal cognitive ability. We analyzed models that made different predictions depending on if effects were genuinely increasing or spurious due to residual confounding and regression to the mean. If findings converge, the case for an increasing effect of breastfeeding on the child's cognitive ability and educational outcomes is strengthened. If, on the other hand, findings do not converge, conclusions about a genuine increasing effect may be challenged.

## Methods

### Study population and measures

We refer to Pereyra-Elìas et al. [15,16] for more comprehensive information on the study population, used measurements, descriptive statistics, etc. In short, the UK Millennium Cohort Study [33–39] is a nationally representative study that recruited 18,552 families with children born at the turn of the new century. Data has been collected at sweeps at ages 9 months and 3, 5, 7, 11, 14, and 17 years [40]. Mothers were asked how long the child had been breastfed at the sweeps at ages 9 months, 3 years, and 5 years. At sweep 6, when the children were approximately 14 years old, mothers completed a test of word comprehension. Following Pereyra-Elìas et al. [15,16], we used this test score as a measure of the mother's cognitive ability. In the present study, the analytic sample size varied between 5797 (grades in English) and 9409 (verbal ability at age 5) with a mean value of 8482. These analytic sample sizes could probably be labeled as large, which is advantageous in that it contributes to high statistical power in the analyses.

The children's verbal ability was assessed with the British Ability Scales (BAS) subtests Naming Vocabulary (age 5), Word Reading (age 7), and Verbal Similarities (age 11). At age 14, the children completed a test of word comprehension (similar to the test completed by their mothers at the same sweep). The children's spatial ability was assessed with the BAS subtest Pattern Construction (ages 5 and 7). At age 11, the children completed the Cambridge Neuropsychological Test Automated Battery (CANTAB). Following Pereyra-Elìas et al. [15], we analyzed the Strategy (assessment of how systematically the tests were executed) and Number or Errors aspects of the CANTAB. The latter two scores were reversed so high scores would

indicate high ability. Grades in mathematics and in English were measured at age 16 (sweep 7). For increased comparability, all ability scores and grades were standardized ($M = 0$, $SD = 1$). Breastfeeding was coded as years of breastfeeding duration.

## Ethics statement

All sweeps of the UK Millennium Cohort Study have been approved by the National Health Services (NHS) Research Ethics Committee (MREC). Written informed consent has been obtained from parents, as well as from the children themselves as they have grown up [40]. For the present study, we accessed anonymized data on 26/6/2023 from the UK Data Service platform (data are available at http://doi.org/10.5255/UKDA-SN-8172-4). No ethical approval was required for this secondary analysis.

## Statistical analyses and predictions

Effects of breastfeeding on the eight (four verbal and four spatial) measures of children's cognitive ability and two grades when adjusting for maternal cognitive ability were estimated with ordinary least squares regression. Here, both a hypothesis of a genuine increasing effect and a hypothesis of a spurious effect due to residual confounding predicted positive adjusted effects. Effects of breastfeeding on maternal ability when adjusting for the child's ability and educational outcomes were also estimated with ordinary least squares regression. Here, a hypothesis of a genuine increasing effect predicted negative adjusted effects. This would mean that among equally able children, those who had been breastfed more had less able mothers but had still been able to "catch up" with their less breastfed peers with more able mothers. As an example, if among children with average ability (standardized score = 0), those who had been breastfed more had less able mothers (e.g. average maternal score = -0.5) than those who had been breastfed less (e.g. average maternal score = 0.5), this would mean that the intergenerational change in ability had been more positive for the children who had been breastfed more (0 – (-0.5) = 0.5) compared with those who had been breastfed less (0–0.5 = -0.5). A negative effect of breastfeeding on maternal cognitive ability when adjusting for the child's ability would indicate that breastfeeding had compensated for having a less able mother. Contrarily, due to the positive associations between breastfeeding and maternal and the child's ability and educational outcomes, a hypothesis of spuriousness due to residual confounding predicted a positive effect of breastfeeding on maternal ability when adjusting for the child's ability and educational outcomes. It should be noted that we do not claim a causal effect of children's cognitive ability on maternal cognitive ability. The reason for estimating the effect of breastfeeding on maternal cognitive ability while adjusting for the child's ability, in addition to estimating the effect of breastfeeding on the child's ability while adjusting for maternal ability, was to assess if the latter effect was truly increasing or spurious due to residual confounding.

In addition to ordinary least squares regression, we used latent change score modeling [41–43] to estimate effects of breastfeeding on intergenerational change in cognitive ability from mothers to their children (Fig 1). In these structural equation models, the child's ability and educational outcomes were regressed on maternal ability and a latent intergenerational change score, with both regression effects set to unity. This means that the child's ability and educational outcomes were defined as maternal ability + change and, consequently, that change was defined as the child's ability–maternal ability. The latent change score was, in turn, regressed on breastfeeding. According to a hypothesis of a genuine increasing effect, this latter effect should be positive. Fitting the three models described above on the same data allowed us to evaluate if breastfeeding appears to have a truly increasing effect on children's cognitive ability or whether the effect is spurious due to residual confounding and regression to the mean.

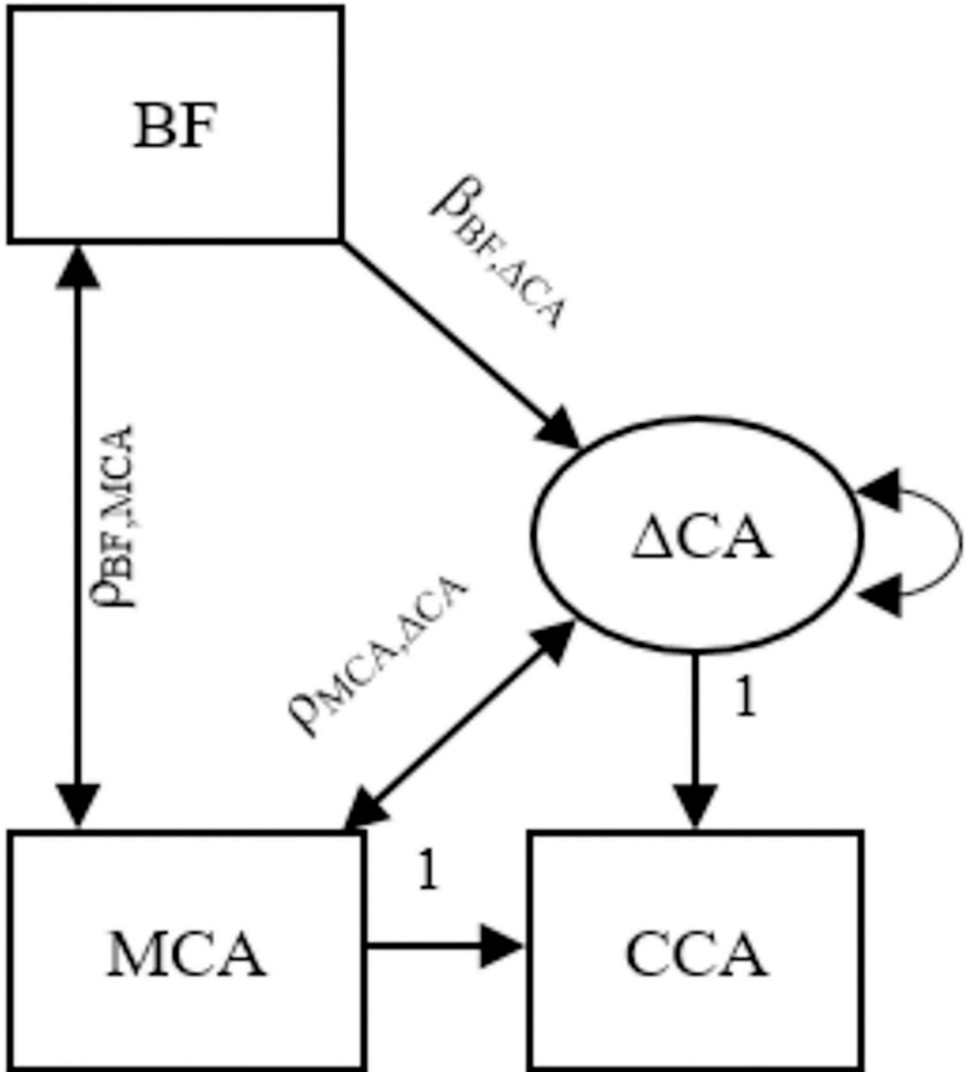

**Fig 1. Latent change score modeling.** Latent change score modeling where the child's cognitive ability (CCA) was regressed on maternal cognitive ability (MCA) and intergenerational change in cognitive ability (ΔCA). Intergenerational change in ability was, in turn, regressed on breastfeeding (BF).

Three differences between the present analyses and the analyses conducted by Pereyra-Elìas et al. [15,16] can be noted: (1) Pereyra-Elìas et al. used breastfeeding as a categorical predictor and compared the effect of categorized breastfeeding durations (< 2 months, 2 to < 4 months, 4 to < 6 months, 6 to < 12 months, and ≥ 12 months, respectively) with "never breastfed" as the reference category. We, on the other hand, used breastfeeding as a continuous predictor with values coded as years of breastfeeding; (2) Pereyra-Elìas et al. [16] predicted"failure" and "high pass" as dichotomous educational outcomes in mathematics and in English, respectively. We, on the other hand, used grades on a scale from 1 to 9 (then standardized) as continuous outcomes; (3) We did not adjust for possible confounders in addition to maternal cognitive ability while Pereyra-Elìas et al. adjusted for a large number of potential confounders, including socioeconomic position, maternal age and use of alcohol and tobacco, etc. This difference reflects the difference in the objectives of the studies. Pereyra-Elìas et al. wanted to evaluate if the effect of breastfeeding on the child's cognitive ability and educational outcomes persisted

when adjusting for a large number of potential confounders. Our objective, on the other hand, was to assess if findings from different models, but all with the same variables (breastfeeding, maternal cognitive ability, and the child's cognitive ability and educational outcomes), would converge. As the models were fitted to the same data, characteristics of the sample were constant across the analyzed models and could, consequently, not explain possible differences in what the models indicate. For example, as the analyzed data were the same, the mothers' socio-economic position would not be able to explain if, for example, one model would indicate an increasing effect of breastfeeding on the child's cognitive ability while another model would indicate a decreasing effect. If we find that effects of breastfeeding on the child's cognitive ability appear to be due to residual confounding due to measurement error in the confounder, adding additional potential confounders to the model would correspond to assessing if residual confounding is confounded by other confounders. We believe that this would complicate interpretations of findings with little or no added value.

It is important to note that the present study was not meant as an exact replication of Pereyra-Elìas et al. [15,16]. Instead, we set out to evaluate if effects of breastfeeding on the child's cognitive ability and educational outcomes when adjusting for maternal cognitive ability appear to be genuine (i.e. non-spurious) or spurious due to residual confounding due to measurement error. The evaluation was conducted by assessing the convergence of findings from different models. However, as Pereyra-Elìas et al. analyzed effects between breastfeeding and cognitive ability and educational outcomes in the same dataset, it makes sense to compare our findings to theirs. Analyses were conducted with R 4.1.3 statistical software [44] employing the lavaan package [45]. The analytic script is available at the Open Science Framework at https://osf.io/c93ha/.

## Results

Bivariate correlations between study variables are presented in Table 1. Mirroring findings by Pereyra-Elìas et al. [15,16], we found positive effects of breastfeeding on the child's cognitive ability and educational outcomes when adjusting for maternal cognitive ability (Table 2, column 1). For example, when adjusting for maternal cognitive ability, for an increase in breastfeeding by one year the child's verbal ability at age 14 was predicted to increase by 0.192 standard deviations (Fig 2, panel A). This could be seen to indicate that breastfeeding had an increasing effect on the child's cognitive ability and educational outcomes.

**Table 1. Bivariate correlations between study variables.**

| Variable | 2. | 3. | 4. | 5. | 6. | 7. | 8. | 9. | 10. | 11. | 12. |
|---|---|---|---|---|---|---|---|---|---|---|---|
| 1.Mat. cog. ab. | 0.23 | 0.40 | 0.30 | 0.28 | 0.34 | 0.21 | 0.26 | 0.16 | 0.21 | 0.26 | 0.26 |
| 2.Breastfeeding | | 0.08 | 0.15 | 0.13 | 0.17 | 0.07 | 0.09 | 0.07 | 0.10 | 0.14 | 0.13 |
| 3.Verbal, 5 | | | 0.35 | 0.40 | 0.34 | 0.45 | 0.33 | 0.13 | 0.20 | 0.26 | 0.24 |
| 4.Verbal, 7 | | | | 0.33 | 0.39 | 0.33 | 0.32 | 0.16 | 0.25 | 0.37 | 0.36 |
| 5.Verbal, 11 | | | | | 0.34 | 0.29 | 0.28 | 0.14 | 0.19 | 0.33 | 0.31 |
| 6.Verbal, 14 | | | | | | 0.23 | 0.27 | 0.16 | 0.23 | 0.34 | 0.34 |
| 7.Spatial, 5 | | | | | | | 0.54 | 0.18 | 0.30 | 0.32 | 0.21 |
| 8.Spatial, 7 | | | | | | | | 0.22 | 0.35 | 0.41 | 0.26 |
| 9.Spatial, 11 (S) | | | | | | | | | 0.65 | 0.21 | 0.12 |
| 10.Spatial, 11 (E) | | | | | | | | | | 0.30 | 0.20 |
| 11.Grades, Math | | | | | | | | | | | 0.66 |
| 12.Grades, Eng. | | | | | | | | | | | |

Note: Mat. cog. ab. = maternal cognitive ability, numbers 5, 7, 11, and 14 refer to the age of the child at the time of measurement, S = strategy, E = errors, Eng. = English, all correlations are highly significant ($p < 0.001$).

**Table 2. Effects (and 95% CI) of breastfeeding on the child's verbal and spatial cognitive ability and grades at different ages when adjusting for maternal cognitive ability, and vice versa, as well as on the intergenerational change in cognitive ability.**

| CCA | β(BF,CCA.MCA) | β(BF,MCA.CCA) | β(BF,ΔCA) |
|---|---|---|---|
| Verbal | | | |
| Age 5 | 0.001 (-0.035; 0.038) | 0.384 (0.348; 0.421)* | -0.292 (-0.338; -0.245)* |
| Age 7 | 0.151 (0.111; 0.192)* | 0.398 (0.358; 0.437)* | -0.177 (-0.229; -0.125)* |
| Age 11 | 0.138 (0.098; 0.178)* | 0.407 (0.368; 0.447)* | -0.201 (-0.253; -0.150)* |
| Age 14 | 0.192 (0.152; 0.233)* | 0.365 (0.326; 0.404)* | -0.136 (-0.185; -0.087)* |
| Spatial | | | |
| Age 5 | 0.055 (0.016; 0.094)† | 0.424 (0.386; 0.463)* | -0.321 (-0.373; -0.268)* |
| Age 7 | 0.087 (0.046; 0.128)* | 0.433 (0.393; 0.472)* | -0.280 (-0.332; -0.228)* |
| Age 11 (S) | 0.092 (0.050; 0.135)* | 0.450 (0.410; 0.491)* | -0.326 (-0.384; -0.268)* |
| Age 11 (E) | 0.120 (0.078; 0.162)* | 0.433 (0.393; 0.473)* | -0.264 (-0.318; -0.210)* |
| Grades | | | |
| Mathematics | 0.175 (0.125; 0.225)* | 0.386 (0.337; 0.434)* | -0.184 (-0.244; -0.124)* |
| English | 0.158 (0.107; 0.210)* | 0.399 (0.349; 0.448)* | -0.207 (-0.268; -0.147)* |

Note: CCA = the child's cognitive ability, BF = breastfeeding, MCA = maternal cognitive ability,

ΔCA = intergenerational change in cognitive ability from mother to child, S = strategy, E = errors

* $p < 0.001$

† $p < 0.01$.

However, effects of breastfeeding on maternal cognitive ability when adjusting for the child's cognitive ability and educational outcomes were also positive and statistically significant (Table 2, column 2). This indicated that among equally able children, those who had been breastfed more had also more able mothers and had, consequently, experienced a more negative intergenerational change in ability compared with equally able children who had been breastfed less (Fig 2, panel B). The negative association between breastfeeding and intergenerational change was also seen in analyses with latent change score modeling (Table 2, column 3, and Fig 2, panel C).

As data contained some extreme values, especially on breastfeeding, we calculated nonparametric bootstrapped estimates. Except for the effect of breastfeeding on the child's verbal ability at age 5 when adjusting for maternal cognitive ability, all effects in Table 2 were verified as statistically significant and to have the same sign (positive or negative) as in Table 2.

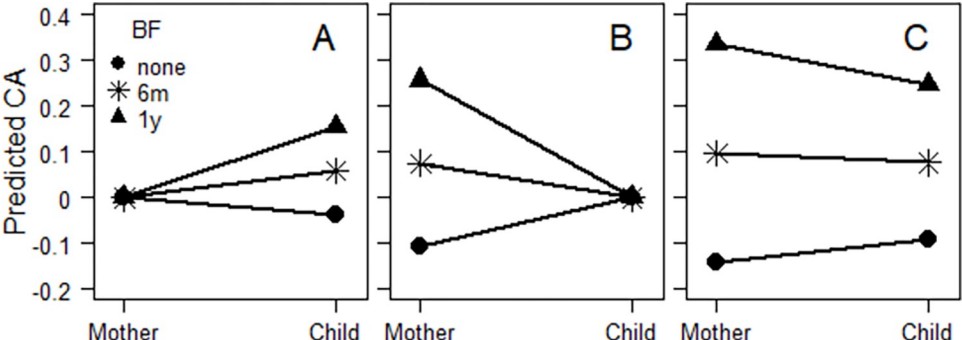

**Fig 2. Predicted cognitive ability.** Maternal and the child's predicted cognitive ability (verbal ability at 14 years of age for the child) as functions of breastfeeding (BF, with the levels none, 6 months, and 1 year), separately for when conditioning on average maternal cognitive ability (A), average child ability (B), and when not conditioning (C).

## Discussion

We set out to reanalyze the data used by Pereyra-Elías et al. [15,16] and to evaluate their conclusion that "the role of breastfeeding on the child's cognitive scores should not be underestimated" ([15], p. 15) and their recommendation that breastfeeding should be encouraged. The present findings show that attempts to draw such a conclusion leads to incongruous results. The results as shown in Table 2 and Fig 2 would yield contradictory conclusions as follows:

1. Mothers should breastfeed their children if they want them to have higher cognitive ability and educational outcomes than the non-breastfed children of mothers who have the same cognitive ability as themselves (Fig 2, panel A);

2. However, mothers should not breastfeed their children if they want them to have the same cognitive ability and educational outcomes as the breastfed children of mothers who have higher cognitive ability than themselves. By not breastfeeding their children, mothers can compensate for their own low cognitive ability (Fig 2, panel B);

3. Furthermore, mothers should not breastfeed their children if they want them to have higher ability than themselves (Fig 2, panel C).

Obviously, these conclusions, although grounded in empirical findings, would be contradictory. Instead, we suggest that the associations may have been spurious due to residual confounding and regression to the mean rather than due to genuine effects of breastfeeding. Breastfeeding mothers tend to have higher cognitive ability than their non-breastfeeding counterparts. Therefore, we may suspect that among mothers with the same measured cognitive ability, those who breastfeed their children have, on average, higher true cognitive ability than mothers with the same measured cognitive ability but who do not breastfeed their children. This would mean a more negative residual in the measurement of cognitive ability among breastfeeding compared with non-breastfeeding mothers with the same measured cognitive ability. Children probably tend to inherit (partly) their parents' true rather than their measured cognitive ability. Consequently, even if adjusting for maternal cognitive ability, we should expect a positive, but spurious, effect of breastfeeding on the child's cognitive ability and educational outcomes even if breastfeeding would have no genuine effect.

The main general point of the present study is that it is difficult to adjust for confounders that have been measured with error, i.e. with less than perfect reliability. One way to assess if measurement error may be a problem is to estimate, as we did in the present study, the effect of the predictor X on the possible confounder Z while adjusting for the outcome Y in addition to the effect of X on Y when adjusting for Z [27,46]. If interpretations of the two models converge, the case that X has an effect on Y over and above the influence of Z is strengthened. However, in the present study we found no convergence. A positive effect of breastfeeding on the child's cognitive ability and educational outcomes when adjusting for maternal cognitive ability may appear to have suggested a genuine positive impact of breastfeeding. However, the effect of breastfeeding on maternal ability when adjusting for the child's ability and educational outcomes was also positive (and stronger than the adjusted effect on the child's ability). This suggested, contrarily, that less breastfed children had, as a manner of speaking, catched up with their more breastfed peers although being dealt a worse hand at the outset of life (in the form of less able mothers). The latent change score models also indicated lack of positive impact of breastfeeding on intergenerational change in cognitive ability from mothers to their children. Due to these non-converging findings, claims about a genuine positive effect of breastfeeding on the child's cognitive ability and educational outcomes, by Pereyra-Elías et al. [15,16] and others, may be challenged.

## Limitations

The present study, as well as the original studies by Pereyra-Elìas et al. [15,16], was based on data from a modern Western society (the UK). We concluded that estimated effects of breastfeeding on the child's cognitive ability and educational outcomes when adjusting for maternal cognitive ability may have been spurious due to residual confounding and regression to the mean rather than genuine. However, it is possible that this conclusion does not apply to other circumstances and societies, e.g. with nutritional deficiency.

As described in the Method section, we did not adjust for potential confounders in addition to maternal cognitive ability, e.g. socioeconomic position. We made this choice in order not to distract from the main point, i.e. that analyses can suggest very different conditions depending on which model is fitted to data. It is important to bear in mind that potential confounders were constant across the analyzed models and could not, consequently, account for the inconsistent findings. For example, as data came from the same mothers and children, the mothers' socioeconomic position could not explain why some models suggested an increasing effect of breastfeeding on the child's cognitive ability and educational outcomes while other models indicated a decreasing effect.

Maternal cognitive ability was measured after all but the last of the measurements of their child's cognitive ability, at age 14. Hence, it is possible, at least theoretically, that the child's cognitive ability may have affected maternal cognitive ability rather than, as assumed, the other way around. However, this should not affect our main conclusion that estimated effects of breastfeeding of the child's cognitive ability when adjusting for maternal cognitive ability could be due to residual confounding due to error in the measurement of maternal ability. If anything, this suboptimal timing of measurements should make us even more cautious not to assume a genuine increasing effect of breastfeeding.

We have, in a manner of speaking, retained the null hypothesis of no genuine increasing effect of breastfeeding on the child's cognitive ability and educational outcomes, and retention of a null hypothesis could be due to lack of statistical power. However, lack of power is usually seen as a potential problem when findings are not statistically significant. In the present case, with a single exception, all estimated effects were highly statistically significant (see Table 2). However, the effects had the wrong sign compared with what they would be expected to have with a genuine increasing effect. If power were increased, for example by increasing the sample size, the estimated effects would be expected to become even more statistically significant but not to change sign. Consequently, it is hard to see how lack of agreement between the present findings and a hypothesis of a genuine increasing effect could be explained by lack of power.

The objective of the current study was not to present a comprehensive review of research on cognitive ability and its determinants. Instead, and more specifically, the objective was to reanalyze the data used by Pereyra-Elìas et al. [15,16] and to evaluate their conclusion that a positive effect of breastfeeding on the child's cognitive ability and educational outcomes persisted when adjusting for possible confounders, including maternal cognitive ability. Readers interested in comprehensive reviews are recommended to read, for example, works by Deary [e.g. 47,48] or Sternberg [e.g. 49].

## Conclusions

Pereyra-Elìas et al. [15,16] found positive effects of breastfeeding on the child's cognitive ability and educational outcomes even when adjusting for maternal cognitive ability in addition to a large number of other potential confounders, e.g. socioeconomic position. Pereyra-Elìas et al. concluded that "the role of breastfeeding on the child's cognitive scores should not be underestimated" ([15], p. 15). However, in the present reanalyses of the same data, we found

incongruent effects indicating simultaneous increasing and decreasing effects of breastfeeding on the child's cognitive ability and educational outcomes. We conclude that the finding by Pereyra-Elìas et al. may have been due to residual confounding due to error in the measurement of maternal cognitive ability. Consequently, it appears premature to assume a genuine increasing effect of breastfeeding on the child's cognitive ability and educational outcomes and claims in this regard, by Pereyra-Elìas et al. and others, may be challenged.

## Acknowledgments

We are grateful to the Centre for Longitudinal Studies (CLS), UCL Social Research Institute, for the use of these data and to the UK Data Service for making them available. However, neither CLS nor the UK Data Service bear any responsibility for the analysis or interpretation of these data.

## Author Contributions

**Conceptualization:** Kimmo Sorjonen, Gustav Nilsonne, Michael Ingre, Bo Melin.

**Data curation:** Kimmo Sorjonen.

**Formal analysis:** Kimmo Sorjonen.

**Investigation:** Kimmo Sorjonen, Gustav Nilsonne, Michael Ingre, Bo Melin.

**Methodology:** Kimmo Sorjonen.

**Project administration:** Kimmo Sorjonen.

**Supervision:** Gustav Nilsonne, Michael Ingre, Bo Melin.

**Visualization:** Kimmo Sorjonen.

**Writing – original draft:** Kimmo Sorjonen.

**Writing – review & editing:** Gustav Nilsonne, Michael Ingre, Bo Melin.

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
