## [Decision Letter · Decision Letter 0]

14 Nov 2023

PONE-D-23-20511Breastfeeding, cognitive ability, and residual confounding: A comment on studies by Pereyra-Elìas et al.PLOS ONE

Dear Dr. Sorjonen,

Thank you for submitting your manuscript to PLOS ONE. After careful consideration, we feel that it has merit but does not fully meet PLOS ONE’s publication criteria as it currently stands. Therefore, we invite you to submit a revised version of the manuscript that addresses the points raised during the review process.

We look forward to receiving your revised manuscript.

Kind regards,

Rita Amiel Castro

Academic Editor

PLOS ONE

Journal Requirements:

Reviewers' comments:

Reviewer's Responses to Questions

**Comments to the Author**

1. Is the manuscript technically sound, and do the data support the conclusions?

Reviewer #1: Yes

Reviewer #2: Yes

Reviewer #3: Yes

2. Has the statistical analysis been performed appropriately and rigorously? 

Reviewer #1: Yes

Reviewer #2: Yes

Reviewer #3: Yes

3. Have the authors made all data underlying the findings in their manuscript fully available?

Reviewer #1: No

Reviewer #2: Yes

Reviewer #3: Yes

4. Is the manuscript presented in an intelligible fashion and written in standard English?

Reviewer #1: Yes

Reviewer #2: Yes

Reviewer #3: Yes

5. Review Comments to the Author

Reviewer #1: This article reanalyzes the data used by Pereyra-Elìas et al. to "evaluate their conclusion that a positive effect of breastfeeding on the child’s cognitive ability and educational outcomes persisted when adjusting for possible confounders, including maternal cognitive ability." It argues that "Although Pereyra-Elìas et al. adjusted for a large number of possible confounders, e.g. socioeconomic position and maternal cognitive ability, they did not appear to consider or discuss the possible impact of measurement error." Consequently, this article focuses on measurement error and regression to the mean. While the focus is legitimate, I think this is also the weakness of this study. Both Pereyra-Elìas et al. and the authors of the current paper ignored the role schools/teachers may play on the cognitive development of children. This paper did not include any confounders in the models. The authors could investigate if any missing confounders (e.g., the variation of outcomes by schools or school districts) might be the reason for non-spurious relationships. Although cognitive ability is hereditary, it also largely depends on environmental factors and the learning process, and different schools or teachers may help children's cognitive development by adopting different teaching-learning methods and materials. I think the authors need to provide a comprehensive review of literature on the determinants of cognitive ability. If even after controlling for some of these determinants (e.g., schools, mother's cognitive ability, and socioeconomic status) the residual confounding appears to be significant, they could more convincingly conclude that "claims about a genuine positive effect of breastfeeding on the child’s cognitive ability and educational outcomes, by Pereyra-Elìas et al. and others, may be challenged."

There are also some "absurd" statements (lines 254-262) which I think should be omitted to keep the tone of the paper "serious." It is assumed that the contradictory recommendations, although based on empirical findings, do not make any sense from a policy perspective. It is also important to explain if there is any underlying causal mechanism to estimate "the effect of breastfeeding on intergenerational change in cognitive ability...backward from children to their mothers...."

There is a minor issue at line 182 with the punctuation: ”failure”. It needs to be fixed.

In sum, I would expect a more comprehensive literature review on the role schools or learning process may play on children's cognitive ability and the authors include this confounding variable (along with others) in their models. Otherwise, the main claim about a spurious relationship remains problematic.

Reviewer #2: Rationale and statement of the problem, particularly an argument is not strong enough. So, it would better if they can revise by focus on this main research problem. Importantly, they need to clarify research methodology. For example, they have highly number of sample size. So that, it would improve if they can give some more explantion of some advanctages when using this amount for their analysis. Besides, they have to discuss and give some reason when using this analytical approach. At the same time, transition between paragraph and tables that related are not quite related. Therefore, they can revise a bit.

Reviewer #3: Perhaps you could mention the comparatively low parent-child correlation for ability (I think it was reading) of 0.35, that is Sullivan’s (2021, p. 20) estimate. Generally, the mother or parent-child CA correlation is higher at 0.41 or 0.42 (Bouchard & McGue, 1981; Daniels, Devlin, & Roeder, 1997, p. 56; Plomin, DeFries, Knopik, & Neiderhiser, 2013, p. 195). I am not sure what was the mother-child correlation for the measured used by Pereyra-Elías, Quigley and Carson {, 2022 #6845} or in your analysis.

I think it is worth noting that that the Millenium cohort mother-child correlation is lower than other estimates. Please include your bivariate correlation. Also, the scores are vocabulary scores, not IQ scores and from a shortened version so more error. These ideas support your argument.

Bouchard, T. J., & McGue, M. (1981). Familial studies of intelligence: A review. Science, 212(4498), 1055-1059. doi:https://doi.org/10.1126/science.7195071.

Daniels, M., Devlin, B., & Roeder, K. (1997). Of genes and IQ. In B. Devlin, S. E. Fienberg, D. P. Resnick, & K. Roeder (Eds.), Intelligence, genes, and success: Scientists respond to the Bell Curve (pp. 45-70). New York: Springer-Verlag.

Pereyra-Elías, R., Quigley, M. A., & Carson, C. (2022). To what extent does confounding explain the association between breastfeeding duration and cognitive development up to age 14? Findings from the UK millennium cohort study. PLOS ONE, 17(5), e0267326. doi:10.1371/journal.pone.0267326.

Plomin, R., DeFries, J. D., Knopik, V. S., & Neiderhiser, J. M. (2013). Behavioral genetics (6th ed.). New York: Worth Publishers.

Sullivan, A., Moulton, V., & Fitzsimons, E. (2021). The intergenerational transmission of language skill. The British Journal of Sociology, 72(207–232). doi:https://doi.org/10.1111/1468-4446.12780.

6. PLOS authors have the option to publish the peer review history of their article (what does this mean?). If published, this will include your full peer review and any attached files.

Reviewer #1: **Yes: **Amm Quamruzzaman

Reviewer #2: **Yes: **Yothin Sawangdee

Reviewer #3: **Yes: **Gary N Marks

---

## [Author Response · Author response to Decision Letter 0]

21 Nov 2023

Journal Requirements:

Response: We have tried to follow the requirements to the best of our ability.

Response: We are not allowed to share the data from the UK Millennium Cohort Study. However, as we say in the manuscript (lines 109-111), the data is publicly available from the UK Data Service. We have added a doi-link (line 111). We have also added a data availability statement (lines 345-348). We would like to point out that neither did one of the challenged studies by Pereyra-Elìas et al., using the same dataset and published in PLOS ONE last year, share the data but referred to the UK Data Service. Other studies using the UK Millennium Cohort Study published in PLOS ONE that refer to the UK Data Service without sharing the data include the following:

1. Campbell M, Straatmann VS, Lai ETC, Potier J, Pinto Pereira SM, Wickham SL, et al. Understanding social inequalities in children being bullied: UK Millennium Cohort Study findings. Zeeb H, editor. PLoS ONE. 2019;14: e0217162. doi:10.1371/journal.pone.0217162

2. Masukume G, Khashan AS, Morton SMB, Baker PN, Kenny LC, McCarthy FP. Caesarean section delivery and childhood obesity in a British longitudinal cohort study. Simeoni U, editor. PLoS ONE. 2019;14: e0223856. doi:10.1371/journal.pone.0223856

3. Emmott EH, Mace R. Practical Support from Fathers and Grandmothers Is Associated with Lower Levels of Breastfeeding in the UK Millennium Cohort Study. Raju T, editor. PLoS ONE. 2015;10: e0133547. doi:10.1371/journal.pone.0133547

4. Alterman N, Johnson S, Carson C, Petrou S, Kurinzcuk JJ, Macfarlane A, et al. Gestational age at birth and academic attainment in primary and secondary school in England: Evidence from a national cohort study. Duerden E, editor. PLoS ONE. 2022;17: e0271952. doi:10.1371/journal.pone.0271952

5. Gitsels LA, Cortina-Borja M, Rahi JS. Is amblyopia associated with school readiness and cognitive performance during early schooling? Findings from the Millennium Cohort Study. Awadein A, editor. PLoS ONE. 2020;15: e0234414. doi:10.1371/journal.pone.0234414 

Response: We have used the PLOS ONE reference template in Zotero. We do not cite any retracted papers. We have added four new references, namely references number 11 (line 45) and 47-49 (line 324).

#### Response to a requirement from editor Edrian Nim Tolentino in a separate mail on 2023-11-21 ####

1. In the Methods section please include the informed consent statement to reflect whether "written or verbal" informed consent was obtained from all participants for inclusion in the study.

If the need for consent was waived by the ethics committee, please include this information.

Response: We have added “written” (line 108).

We note that in the study, published in PLOS ONE last year, that we challenge in the present paper, Pereyra-Elías et al. [1] were, differently from us, not required to state anything about consent, although they used the same data (the UK Millennium Cohort Study) as we do in our present study. 

1. Pereyra-Elías R, Quigley MA, Carson C. To what extent does confounding explain the association between breastfeeding duration and cognitive development up to age 14? Findings from the UK Millennium Cohort Study. PLOS ONE. 2022;17: e0267326. doi:10.1371/journal.pone.0267326

####

Reviewers' comments:

Reviewer's Responses to Questions

Comments to the Author

1. Is the manuscript technically sound, and do the data support the conclusions?

Reviewer #1: Yes

Reviewer #2: Yes

Reviewer #3: Yes

2. Has the statistical analysis been performed appropriately and rigorously?

Reviewer #1: Yes

Reviewer #2: Yes

Reviewer #3: Yes

3. Have the authors made all data underlying the findings in their manuscript fully available?

Reviewer #1: No

Reviewer #2: Yes

Reviewer #3: Yes

4. Is the manuscript presented in an intelligible fashion and written in standard English?

Reviewer #1: Yes

Reviewer #2: Yes

Reviewer #3: Yes

5. Review Comments to the Author

Reviewer #1: This article reanalyzes the data used by Pereyra-Elìas et al. to "evaluate their conclusion that a positive effect of breastfeeding on the child’s cognitive ability and educational outcomes persisted when adjusting for possible confounders, including maternal cognitive ability." It argues that "Although Pereyra-Elìas et al. adjusted for a large number of possible confounders, e.g. socioeconomic position and maternal cognitive ability, they did not appear to consider or discuss the possible impact of measurement error." Consequently, this article focuses on measurement error and regression to the mean. While the focus is legitimate, I think this is also the weakness of this study. Both Pereyra-Elìas et al. and the authors of the current paper ignored the role schools/teachers may play on the cognitive development of children. This paper did not include any confounders in the models. The authors could investigate if any missing confounders (e.g., the variation of outcomes by schools or school districts) might be the reason for non-spurious relationships. Although cognitive ability is hereditary, it also largely depends on environmental factors and the learning process, and different schools or teachers may help children's cognitive development by adopting different teaching-learning methods and materials. I think the authors need to provide a comprehensive review of literature on the determinants of cognitive ability. If even after controlling for some of these determinants (e.g., schools, mother's cognitive ability, and socioeconomic status) the residual confounding appears to be significant, they could more convincingly conclude that "claims about a genuine positive effect of breastfeeding on the child’s cognitive ability and educational outcomes, by Pereyra-Elìas et al. and others, may be challenged."

Response: We do not agree that adjusting for schools/teachers is crucial for our conclusions. As we say in the paper (lines 173-182 and 294-299), we compared effects in different models fitted to the same data. Consequently, characteristics of the children (e.g. which school they went to and which teachers they had) and their mothers were constant across the analyzed models. Hence, such characteristics cannot explain the paradoxical findings of simultaneous increasing and decreasing effects of breastfeeding on children’s cognitive ability. As an analogy, even if performance in high jump probably is associated with height, height cannot, because it is constant, explain if a person jumps higher with shoe brand A compared with shoe brand B. 

 Moreover, adjustment for potential confounders is usually carried out in order to evaluate if a discovered association can be due to influence by a confounder, i.e. if the association is spurious. In the present case we found the effect of breastfeeding on children’s cognitive ability to be spurious. To look for confounders that may explain this finding would be like testing if an athlete’s poor performance could be due to use of illicit performance-enhancing drugs or if somebody’s splendid health could be due to iron deficiency. 

 Additionally, we do not believe that information on the children’s schools and teachers are available in data from the UK Millenium Cohort Study. To require adjustment for these factors would probably make most research on cognitive ability impossible.

The present study was not intended as a comprehensive review of the literature on determinants of cognitive ability. Rather, and more specifically, the objective was (lines 70-73):

to reanalyze the data used by Pereyra-Elìas et al. [15,16] and to evaluate their conclusion that a positive effect of breastfeeding on the child’s cognitive ability and educational outcomes persisted when adjusting for possible confounders, including maternal cognitive ability. 

We have added the following under Limitations (lines 318-324):

The objective of the current study was not to present a comprehensive review of research on cognitive ability and its determinants. Instead, and more specifically, the objective was to reanalyze the data used by Pereyra-Elìas et al. [15,16] and to evaluate their conclusion that a positive effect of breastfeeding on the child’s cognitive ability and educational outcomes persisted when adjusting for possible confounders, including maternal cognitive ability. Readers interested in comprehensive reviews are recommended to read, for example, works by Deary [e.g. 47,48] or Sternberg [e.g. 49].

There are also some "absurd" statements (lines 254-262) which I think should be omitted to keep the tone of the paper "serious." It is assumed that the contradictory recommendations, although based on empirical findings, do not make any sense from a policy perspective. It is also important to explain if there is any underlying causal mechanism to estimate "the effect of breastfeeding on intergenerational change in cognitive ability...backward from children to their mothers...."

Response: Actually, the first of these statements is quite common, echoing, for example, claims by Pereyra-Elìas et al. Our point here is that different models would suggest very different recommendations even if applied to the same data, and there is no way to determine which, if any, of the recommendations is “the correct one”. 

Our intention is not that these recommendations should be used in actual policy decisions (that is why we called them “absurd”). We have changed the word “recommendations” to “conclusions” (lines 240 and 252) and the word “absurd” to “contradictory” (line 253). 

We do not claim causal effects backward in time. However, we believe that it is legitimate to estimate effects on initial levels of a variable while adjusting for subsequent levels in order to evaluate if effects may be truly causal or spurious. For example, if marijuana has a truly causal increasing effect on heart rate, we should see a positive effect of dose of marijuana on subsequent heart rate while adjusting for heart rate at baseline. This would mean that among individuals with the same heart rate at baseline, those who were exposed to a higher dose had a higher heart rate at the subsequent measurement compared with individuals with the same initial heart rate but who were exposed to a lower dose of marijuana. However, in the case of a truly causal increasing effect, dose of marijuana should have a negative effect on heart rate at baseline when adjusting for subsequent heart rate. This would mean that among individuals with the same subsequent heart rate, those who were exposed to a higher dose had a lower heart rate at baseline and had, consequently, experienced a larger increase in heart rate from baseline to the subsequent measurement compared with individuals with the same subsequent heart rate but who had been exposed to a lower dose. We believe the latter analysis is justified even without assuming a causal effect backward in time of subsequent heart rate on initial heart rate. Similarly, we believe it is justified to estimate the effect of breastfeeding on maternal cognitive ability while adjusting for children’s cognitive ability, in order to evaluate if the effect of breastfeeding is truly increasing of spurious, even without assuming a causal effect of children’s cognitive ability on maternal cognitive ability. We have added the following (lines 134-139):

It should be noted that we do not claim a causal effect of children’s cognitive ability on maternal cognitive ability. The reason for estimating the effect of breastfeeding on maternal cognitive ability while adjusting for the child’s ability, in addition to estimating the effect of breastfeeding on the child’s ability while adjusting for maternal ability, was to assess if the latter effect was truly increasing or spurious due to residual confounding.

There is a minor issue at line 182 with the punctuation: ”failure”. It needs to be fixed.

Response: We are not completely sure what you mean. However, we have added a full stop as well as the following (line 164): “We, on the other hand, …”.

In sum, I would expect a more comprehensive literature review on the role schools or learning process may play on children's cognitive ability and the authors include this confounding variable (along with others) in their models. Otherwise, the main claim about a spurious relationship remains problematic.

Response: We do not agree that adjusting for schools/teachers is crucial for our conclusions. As we say in the paper (lines 173-182 and 294-299), we compared effects in different models fitted to the same data. Consequently, characteristics of the children (e.g. which school they went to and which teachers they had) and their mothers were constant across the analyzed models. Hence, such characteristics cannot explain the paradoxical findings of simultaneous increasing and decreasing effects of breastfeeding on children’s cognitive ability. As an analogy, even if performance in high jump probably is associated with height, height cannot, because it is constant, explain if a person jumps higher with shoe brand A compared with shoe brand B. Additionally, we do not believe that information on the children’s schools and teachers are available in data from the UK Millenium Cohort Study. To require adjustment for these factors would probably make most research on cognitive ability impossible.

Reviewer #2: Rationale and statement of the problem, particularly an argument is not strong enough. So, it would better if they can revise by focus on this main research problem. Importantly, they need to clarify research methodology. For example, they have highly number of sample size. So that, it would improve if they can give some more explantion of some advanctages when using this amount for their analysis. Besides, they have to discuss and give some reason when using this analytical approach. At the same time, transition between paragraph and tables that related are not quite related. Therefore, they can revise a bit.

Response: As we say in the paper (lines 70-73):

The objective of the present study was to reanalyze the data used by Pereyra-Elìas et al. [15,16] and to evaluate their conclusion that a positive effect of breastfeeding on the child’s cognitive ability and educational outcomes persisted when adjusting for possible confounders, including maternal cognitive ability.

We have added the following (lines 91-93):

These analytic sample sizes could probably be labeled as large, which is advantageous in that it contributes to high statistical power in the analyses.

We have added the following (lines 148-150):

Fitting the three models described above on the same data allowed us to evaluate if breastfeeding appears to have a truly increasing effect on children’s cognitive ability or whether the effect is spurious due to residual confounding and regression to the mean.

We hope the revisions have improved the “flow” of the text.

Reviewer #3: Perhaps you could mention the comparatively low parent-child correlation for ability (I think it was reading) of 0.35, that is Sullivan’s (2021, p. 20) estimate. Generally, the mother or parent-child CA correlation is higher at 0.41 or 0.42 (Bouchard & McGue, 1981; Daniels, Devlin, & Roeder, 1997, p. 56; Plomin, DeFries, Knopik, & Neiderhiser, 2013, p. 195). I am not sure what was the mother-child correlation for the measured used by Pereyra-Elías, Quigley and Carson {, 2022 #6845} or in your analysis.

I think it is worth noting that that the Millenium cohort mother-child correlation is lower than other estimates. Please include your bivariate correlation. Also, the scores are vocabulary scores, not IQ scores and from a shortened version so more error. These ideas support your argument.

Bouchard, T. J., & McGue, M. (1981). Familial studies of intelligence: A review. Science, 212(4498), 1055-1059. doi:https://doi.org/10.1126/science.7195071.

Daniels, M., Devlin, B., & Roeder, K. (1997). Of genes and IQ. In B. Devlin, S. E. Fienberg, D. P. Resnick, & K. Roeder (Eds.), Intelligence, genes, and success: Scientists respond to the Bell Curve (pp. 45-70). New York: Springer-Verlag.

Pereyra-Elías, R., Quigley, M. A., & Carson, C. (2022). To what extent does confounding explain the association between breastfeeding duration and cognitive development up to age 14? Findings from the UK millennium cohort study. PLOS ONE, 17(5), e0267326. doi:10.1371/journal.pone.0267326.

Plomin, R., DeFries, J. D., Knopik, V. S., & Neiderhiser, J. M. (2013). Behavioral genetics (6th ed.). New York: Worth Publishers.

Sullivan, A., Moulton, V., & Fitzsimons, E. (2021). The intergenerational transmission of language skill. The British Journal of Sociology, 72(207–232). doi:https://doi.org/10.1111/1468-4446.12780.

Response: We have added the following (lines 45-46):

Sullivan et al. [11] reported a correlation of 0.35 between maternal and children’s vocabulary scores.

We have also included a new Table 1 with bivariate correlations as well as the following (line 195):

Bivariate correlations between study variables are presented in Table 1.

---

## [Decision Letter · Decision Letter 1]

2 Jan 2024

Breastfeeding, cognitive ability, and residual confounding: A comment on studies by Pereyra-Elìas et al.

PONE-D-23-20511R1

Dear Dr. Sorjonen,

We’re pleased to inform you that your manuscript has been judged scientifically suitable for publication and will be formally accepted for publication once it meets all outstanding technical requirements.

Kind regards,

Rita Amiel Castro

Academic Editor

PLOS ONE

Additional Editor Comments (optional):

Reviewers' comments:

Reviewer's Responses to Questions

**Comments to the Author**

1. If the authors have adequately addressed your comments raised in a previous round of review and you feel that this manuscript is now acceptable for publication, you may indicate that here to bypass the “Comments to the Author” section, enter your conflict of interest statement in the “Confidential to Editor” section, and submit your "Accept" recommendation.

Reviewer #1: All comments have been addressed

Reviewer #2: All comments have been addressed

2. Is the manuscript technically sound, and do the data support the conclusions?

Reviewer #1: Yes

Reviewer #2: Yes

3. Has the statistical analysis been performed appropriately and rigorously? 

Reviewer #1: Yes

Reviewer #2: Yes

4. Have the authors made all data underlying the findings in their manuscript fully available?

Reviewer #1: No

Reviewer #2: Yes

5. Is the manuscript presented in an intelligible fashion and written in standard English?

Reviewer #1: Yes

Reviewer #2: Yes

6. Review Comments to the Author

Reviewer #1: The authors have addressed all my points. They need to carefully proofread the manuscript and follow the submission guideline to meet the journal's standards.

Reviewer #2: This revision is acceptable. I think it is fine. We can distribute under this journal standard. For example, analysis part and its explanation is easy to understand.

7. PLOS authors have the option to publish the peer review history of their article (what does this mean?). If published, this will include your full peer review and any attached files.

Reviewer #1: **Yes: **Amm Quamruzzaman

Reviewer #2: **Yes: **Yothin Sawangdee

---

## [Editor Report · Acceptance letter]

3 Mar 2024

PONE-D-23-20511R1 

PLOS ONE

Dear Dr. Sorjonen, 

I'm pleased to inform you that your manuscript has been deemed suitable for publication in PLOS ONE. Congratulations! Your manuscript is now being handed over to our production team.

Kind regards, 

on behalf of

Dr. Rita Amiel Castro 

Academic Editor

PLOS ONE